# LEARNING TO SUPEROPTIMIZE PROGRAMS

**Rudy Bunel**[1]**, Alban Desmaison**[1]**, M. Pawan Kumar**[1,2] **& Philip H.S. Torr**[1]
[1]Department of Engineering Science - University of Oxford
[2]Alan Turing Institute
Oxford, UK
{rudy,alban,pawan}@robots.ox.ac.uk,philip.torr@eng.ox.ac.uk

**Pushmeet Kohli**
Microsoft Research
Redmond, WA 98052, USA
pkohli@microsoft.com

## ABSTRACT

Code super-optimization is the task of transforming any given program to a more efficient version while preserving its input-output behaviour. In some sense, it is similar to the paraphrase problem from natural language processing where the intention is to change the syntax of an utterance without changing its semantics. Code-optimization has been the subject of years of research that has resulted in the development of rule-based transformation strategies that are used by compilers. More recently, however, a class of stochastic search based methods have been shown to outperform these strategies. This approach involves repeated sampling of modifications to the program from a proposal distribution, which are accepted or rejected based on whether they preserve correctness and the improvement they achieve. These methods, however, neither learn from past behaviour nor do they try to leverage the semantics of the program under consideration. Motivated by this observation, we present a novel learning based approach for code super-optimization. Intuitively, our method works by learning the proposal distribution using unbiased estimators of the gradient of the expected improvement. Experiments on benchmarks comprising of automatically generated as well as existing ("Hacker's Delight") programs show that the proposed method is able to significantly outperform state of the art approaches for code super-optimization.

## 1 INTRODUCTION

Considering the importance of computing to human society, it is not surprising that a very large body of research has gone into the study of the syntax and semantics of programs and programming languages. Code super-optimization is an extremely important problem in this context. Given a program or a snippet of source-code, super-optimization is the task of transforming it to a version that has the same input-output behaviour but can be executed on a target compute architecture more efficiently. Superoptimization provides a natural benchmark for evaluating representations of programs. As a task, it requires the decoupling of the semantics of the program from its superfluous properties, the exact implementation. In some sense, it is the natural analogue of the paraphrase problem in natural language processing where we want to change syntax without changing semantics.

Decades of research has been done on the problem of code optimization resulting in the development of sophisticated rule-based transformation strategies that are used in compilers to allow them to perform code optimization. While modern compilers implement a large set of rewrite rules and are able to achieve impressive speed-ups, they fail to offer any guarantee of optimality, thus leaving room for further improvement. An alternative approach is to search over the space of all possible programs that are equivalent to the compiler output, and select the one that is the most efficient. If the search is carried out in a brute-force manner, we are guaranteed to achieve super-optimization. However, this approach quickly becomes computationally infeasible as the number of instructions and the length of the program grows.

In order to efficiently perform super-optimization, recent approaches have started to use a stochastic search procedure, inspired by Markov Chain Monte Carlo (MCMC) sampling (Schkufza et al., 2013). Briefly, the search starts at an initial program, such as the compiler output. It iteratively suggests modifications to the program, where the probability of a modification is encoded in a proposal distribution. The modification is either accepted or rejected with a probability that is dependent on the improvement achieved. Under certain conditions on the proposal distribution, the above procedure can be shown, in the limit, to sample from a distribution over programs, where the probability of a program is related to its quality. In other words, the more efficient a program, the more times it is encountered, thereby enabling super-optimization. Using this approach, high-quality implementations of real programs such as the Montgomery multiplication kernel from the OpenSSL library were discovered. These implementations outperformed the output of the `gcc` compiler and even expert-handwritten assembly code.

One of the main factors that governs the efficiency of the above stochastic search is the choice of the proposal distribution. Surprisingly, the state of the art method, Stoke (Schkufza et al., 2013), employs a proposal distribution that is neither learnt from past behaviour nor does it depend on the syntax or semantics of the program under consideration. We argue that this choice fails to fully exploit the power of stochastic search. For example, consider the case where we are interested in performing bitwise operations, as indicated by the compiler output. In this case, it is more likely that the optimal program will contain bitshifts than floating point opcodes. Yet, Stoke will assign an equal probability of use to both types of opcodes.

In order to alleviate the aforementioned deficiency of Stoke, we build a reinforcement learning framework to estimate the proposal distribution for optimizing the source code under consideration. The score of the distribution is measured as the expected quality of the program obtained via stochastic search. Using training data, which consists of a set of input programs, the parameters are learnt via the REINFORCE algorithm (Williams, 1992). We demonstrate the efficacy of our approach on two datasets. The first is composed of programs from "Hacker's Delight" (Warren, 2002). Due to the limited diversity of the training samples, we show that it is possible to learn a prior distribution (unconditioned on the input program) that outperforms the state of the art. The second dataset contains automatically generated programs that introduce diversity in the training samples. We show that, in this more challenging setting, we can learn a conditional distribution given the initial program that significantly outperforms Stoke.

## 2 RELATED WORKS

**Super-optimization** The earliest approaches for super-optimization relied on brute-force search. By sequentially enumerating all programs in increasing length orders (Granlund & Kenner, 1992; Massalin, 1987), the shortest program meeting the specification is guaranteed to be found. As expected, this approach scales poorly to longer programs or to large instruction sets. The longest reported synthesized program was 12 instructions long, on a restricted instruction set (Massalin, 1987).

Trading off completeness for efficiency, stochastic methods (Schkufza et al., 2013) reduced the number of programs to test by guiding the exploration of the space, using the observed quality of programs encountered as hints. In order to improve the size of solvable instances, Phothilimthana et al. (2016) combined stochastic optimizers with smart enumerative solvers. However, the reliance of stochastic methods on a generic unspecific exploratory policy made the optimization blind to the problem at hand. We propose to tackle this problem by learning the proposal distribution.

**Neural Computing** Similar work was done in the restricted case of finding efficient implementation of computation of value of degree $k$ polynomials (Zaremba et al., 2014). Programs were generated from a grammar, using a learnt policy to prioritise exploration. This particular approach of guided search looks promising to us, and is in spirit similar to our proposal, although applied on a very restricted case.

Another approach to guide the exploration of the space of programs was to make use of the gradients of differentiable relaxation of programs. Bunel et al. (2016) attempted this by simulating program execution using Recurrent Neural Networks. However, this provided no guarantee that the network parameters were going to correspond to real programs. Additionally, this method only had the

possibility of performing local, greedy moves, limiting the scope of possible transformations. On the contrary, our proposed approach operates directly on actual programs and is capable of accepting short-term detrimental moves.

**Learning to optimize** Outside of program optimization, applying learning algorithms to improve optimization procedures, either in terms of results achieved or runtime, is a well studied subject. Doppa et al. (2014) proposed imitation learning based methods to deal with structured output spaces, in a "Learning to search" framework. While this is similar in spirit to stochastic search, our setting differs in the crucial aspect of having a valid cost function instead of searching for one.

More relevant is the recent literature on learning to optimize. Li & Malik (2016) and Andrychowicz et al. (2016) learn how to improve on first-order gradient descent algorithms, making use of neural networks. Our work is similar, as we aim to improve the optimization process. However, as opposed to the gradient descent that they learn on a continuous unconstrained space, our initial algorithm is an MCMC sampler on a discrete domain.

Similarly, training a proposal distribution parameterized by a Neural Network was also proposed by Paige & Wood (2016) to accelerate inference in graphical models. Similar approaches were successfully employed in computer vision problems where data driven proposals allowed to make inference feasible (Jampani et al., 2015; Kulkarni et al., 2015; Zhu et al., 2000). Other approaches to speeding up MCMC inference include the work of Salimans et al. (2015), combining it with Variational inference.

## 3 LEARNING STOCHASTIC SUPER-OPTIMIZATION

### 3.1 STOCHASTIC SEARCH AS A PROGRAM OPTIMIZATION PROCEDURE

Stoke (Schkufza et al., 2013) performs black-box optimization of a cost function on the space of programs, represented as a series of instructions. Each instruction is composed of an opcode, specifying what to execute, and some operands, specifying the corresponding registers. Each given input program $\mathcal{T}$ defines a cost function. For a candidate program $\mathcal{R}$ called *rewrite*, the goal is to optimize the following cost function:

$$\text{cost}\,(\mathcal{R}, \mathcal{T}) = \omega_e \times \text{eq}(\mathcal{R}, \mathcal{T}) + \omega_p \times \text{perf}(\mathcal{R}) \tag{1}$$

The term $\text{eq}(\mathcal{R}; \mathcal{T})$ measures how well the outputs of the rewrite match the outputs of the reference program. This can be obtained either exactly by running a symbolic validator or approximately by running test cases. The term $\text{perf}(\mathcal{R})$ is a measure of the efficiency of the program. In this paper, we consider runtime to be the measure of this efficiency. It can be approximated by the sum of the latency of all the instructions in the program. Alternatively, runtime of the program on some test cases can be used.

To find the optimum of this cost function, Stoke runs an MCMC sampler using the Metropolis (Metropolis et al., 1953) algorithm. This allows us to sample from the probability distribution induced by the cost function:

$$p(\mathcal{R}; \mathcal{T}) = \frac{1}{Z} \exp(-\text{cost}\,(\mathcal{R}, \mathcal{T}))). \tag{2}$$

The sampling is done by proposing random moves from a different proposal distribution:

$$\mathcal{R}' \sim q(\,\cdot\,|\mathcal{R}). \tag{3}$$

The cost of the new modified program is evaluated and an acceptance criterion is computed. This acceptance criterion

$$\alpha(\mathcal{R}, \mathcal{T}) = \min\left(1, \frac{p(\mathcal{R}'; \mathcal{T})}{p(\mathcal{R}; \mathcal{T})}\right), \tag{4}$$

is then used as the parameter of a Bernoulli distribution from which an accept/reject decision is sampled. If the move is accepted, the state of the optimizer is updated to $\mathcal{R}'$. Otherwise, it remains in $\mathcal{R}$.

While the above procedure is only guaranteed to sample from the distribution $p(\,\cdot\,; \mathcal{T})$ in the limit if the proposal distribution $q$ is symmetric ($q(\mathcal{R}'|\mathcal{R}) = q(\mathcal{R}|\mathcal{R}')$ for all $\mathcal{R}, \mathcal{R}'$), it still allows us

to perform efficient hill-climbing for non-symmetric proposal distributions. Moves leading to an improvement are always going to be accepted, while detrimental moves can still be accepted in order to avoid getting stuck in local minima.

## 3.2 LEARNING TO SEARCH

We now describe our approach to improve stochastic search by learning the proposal distribution. We begin our description by defining the learning objective (section 3.2.1), followed by a parameterization of the proposal distribution (section 3.2.2), and finally the reinforcement learning framework to estimate the parameters of the proposal distribution (section 3.2.3).

### 3.2.1 OBJECTIVE FUNCTION

Our goal is to optimize the cost function defined in equation (1). Given a fixed computational budget of $T$ iterations to perform program super-optimization, we want to make moves that lead us to the lowest possible cost. As different programs have different runtimes and therefore different associated costs, we need to perform normalization. As normalized loss function, we use the ratio between the best rewrite found and the cost of the initial unoptimized program $\mathcal{R}_0$. Formally, the loss for a set of rewrites $\{\mathcal{R}_t\}_{t=0..T}$ is defined as follows:

$$r(\{\mathcal{R}_t\}_{t=0..T}) = \left( \frac{\min_{t=0..T} \text{cost}\,(\mathcal{R}_t, \mathcal{T})}{\text{cost}\,(\mathcal{R}_0, \mathcal{T})} \right). \tag{5}$$

Recall that our goal is to learn a proposal distribution. Given that our optimization procedure is stochastic, we will need to consider the expected cost as our loss. This expected loss is a function of the parameters $\theta$ of our parametric proposal distribution $q_\theta$:

$$\mathcal{L}(\theta) = \mathbb{E}_{\{\mathcal{R}_t\} \sim q_\theta} \left[ r(\{\mathcal{R}_t\}_{t=0..T}) \right]. \tag{6}$$

### 3.2.2 PARAMETERIZATION OF THE MOVE PROPOSAL DISTRIBUTION

The proposal distribution (3) originally used in Stoke (Schkufza et al., 2013) takes the form of a hierarchical model. The type of the move is initially sampled from a probability distribution. Additional samples are drawn to specify, for example, the affected location in the programs ,the new operands or opcode to use. Which of these probability distribution get sampled depends on the type of move that was first sampled. The detailed structure of the proposal probability distribution can be found in Appendix B.

Stoke uses uniform distributions for each of the elementary probability distributions the model samples from. This corresponds to a specific instantiation of the general stochastic search paradigm. In this work, we propose to learn those probability distributions so as to maximize the probability of reaching the best programs. The rest of the optimization scheme remains similar to the one of Schkufza et al. (2013).

Our chosen parameterization of $q$ is to keep the hierarchical structure of the original work of Schkufza et al. (2013), as detailed in Appendix B, and parameterize all the elementary probability distributions (over the positions in the programs, the instructions to propose or the arguments) independently. The set $\theta$ of parameters for $q_\theta$ will thus contain a set of parameters for each elementary probability distributions. A fixed proposal distribution is kept through the optimization of a given program, so the proposal distribution needs to be evaluated only once, at the beginning of the optimization and not at every iteration of MCMC.

The stochastic computation graph corresponding to a run of the Metropolis algorithm is given in Figure 1. We have assumed the operation of evaluating the cost of a program to be a deterministic function, as we will not model the randomness of measuring performance.

### 3.2.3 LEARNING THE PROPOSAL DISTRIBUTION

In order to learn the proposal distribution, we will use stochastic gradient descent on our loss function (6). We obtain the first order derivatives with regards to our proposal distribution parameters using the REINFORCE (Williams, 1992) estimator, also known as the likelihood ratio estimator (Glynn, 1990) or the score function estimator (Fu, 2006). This estimator relies on a rewriting of

the gradient of the expectation. For an expectation with regards to a probability distribution $x \sim f_\theta$, the REINFORCE estimator is:

$$\nabla_\theta \sum_x f(x; \theta) r(x) = \sum_x r(x) \nabla_\theta f(x; \theta) = \sum_x f(x; \theta) r(x) \nabla_\theta \log(f(x; \theta)), \qquad (7)$$

and provides an unbiased estimate of the gradient.

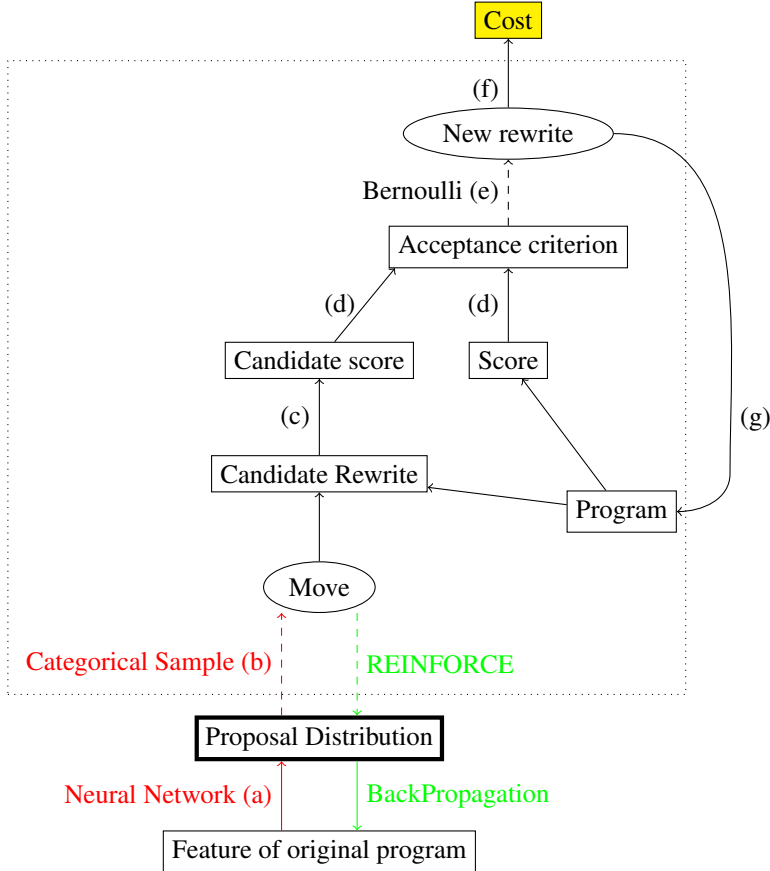

Figure 1: *Stochastic computation graph of the Metropolis algorithm used for program super-optimization. Round nodes are stochastic nodes and square ones are deterministic. Red arrows corresponds to computation done in the forward pass that needs to be learned while green arrows correspond to the backward pass. Full arrows represent deterministic computation and dashed arrows represent stochastic ones. The different steps of the forward pass are:*
*(a) Based on features of the reference program, the proposal distribution q is computed.*
*(b) A random move is sampled from the proposal distribution.*
*(c) The score of the proposed rewrite is experimentally measured.*
*(d) The acceptance criterion (4) for the move is computed.*
*(e) The move is accepted with a probability equal to the acceptance criterion.*
*(f) The cost is observed, corresponding to the best program obtained during the search.*
*(g) Moves b to f are repeated T times.*

A helpful way to derive the gradients is to consider the execution traces of the search procedure under the formalism of stochastic computation graphs (Schulman et al., 2015). We introduce one "cost node" in the computation graphs at the end of each iteration of the sampler. The associated cost corresponds to the normalized difference between the best rewrite so far and the current rewrite after this step:

$$c_t = \min\left(0, \left(\frac{\text{cost}\left(\mathcal{R}_t, \mathcal{T}\right) - \min_{i=0..t-1} \text{cost}\left(\mathcal{R}_i, \mathcal{T}\right)}{\text{cost}\left(\mathcal{R}_0, \mathcal{T}\right)}\right)\right). \qquad (8)$$

The sum of all the cost nodes corresponds to the sum of all the improvements made when a new lowest cost was achieved. It can be shown that up to a constant term, this is equivalent to our objective function (5). As opposed to considering only a final cost node at the end of the $T$ iterations, this has the advantage that moves which were not responsible for the improvements would not get assigned any credit.

For each round of MCMC, the gradient with regards to the proposal distribution is computed using the REINFORCE estimator which is equal to

$$\widehat{\nabla_{\theta,i}}\mathcal{L}(\theta) = (\nabla_\theta \log q_\theta(\mathcal{R}_i|\mathcal{R}_{i-1})) \sum_{t>i} c_t. \tag{9}$$

As our proposal distribution remains fixed for the duration of a program optimization, these gradients needs to be summed over all the iterations to obtain the total contribution to the proposal distribution. Once this gradient is estimated, it becomes possible to run standard back-propagation with regards to the features on which the proposal distribution is based on, so as to learn the appropriate feature representation.

# 4 EXPERIMENTS

## 4.1 SETUP

**Implementation**    Our system is built on top of the Stoke super-optimizer from Schkufza et al. (2013). We instrumented the implementation of the Metropolis algorithm to allow sampling from parameterized proposal distributions instead of the uniform distributions previously used. Because the proposal distribution is only evaluated once per program optimisation, the impact on the optimization throughput is low, as indicated in Table 3.

Our implementation also keeps track of the traces through the stochastic graph. Using the traces generated during the optimization, we can compute the estimator of our gradients, implemented using the Torch framework (Collobert et al., 2011).

**Datasets**    We validate the feasibility of our learning approach on two experiments. The first is based on the Hacker's delight (Warren, 2002) corpus, a collection of twenty five bit-manipulation programs, used as benchmark in program synthesis (Gulwani et al., 2011; Jha et al., 2010; Schkufza et al., 2013). Those are short programs, all performing similar types of tasks. Some examples include identifying whether an integer is a power of two from its binary representation, counting the number of bits turned on in a register or computing the maximum of two integers. An exhaustive description of the tasks is given in Appendix C. Our second corpus of programs is automatically generated and is more diverse.

**Models**    The models we are learning are a set of simple elementary probabilities for the categorical distribution over the instructions and over the type of moves to perform. We learn the parameters of each separate distribution jointly, using a Softmax transformation to enforce that they are proper probability distributions. For the types of move where opcodes are chosen from a specific subset, the probabilities of each instruction are appropriately renormalized. We learn two different type of models and compare them with the baseline of uniform proposal distributions equivalent to Stoke.

Our first model, henceforth denoted the bias, is not conditioned on any property of the programs to optimize. By learning this simple proposal distribution, it is only possible to capture a bias in the dataset. This can be understood as an optimal proposal distribution that Stoke should default to.

The second model is a Multi Layer Perceptron (MLP), conditioned on the input program to optimize. For each input program, we generate a Bag-of-Words representation based on the opcodes of the program. This is embedded through a three hidden layer MLP with ReLU activation unit. The proposal distribution over the instructions and over the type of moves are each the result of passing the outputs of this embedding through a linear transformation, followed by a SoftMax.

The optimization is performed by stochastic gradient descent, using the Adam (Kingma & Ba, 2015) optimizer. For each estimate of the gradient, we draw 100 samples for our estimator. The values of the hyperparameters used are given in Appendix A. The number of parameters of each model is given in Table 1.

| Model | # of parameters |
|---|---|
| Uniform | 0 |
| Bias | 2912 |
| MLP | $1.4 \times 10^6$ |

Table 1: Size of the different models compared.
Uniform corresponds to Stoke Schkufza et al. (2013).

| Model | Training | Test |
|---|---|---|
| Uniform | 57.01% | 53.71% |
| Bias | 36.45 % | 31.82 % |
| MLP | 35.96 % | 31.51 % |

Table 2: Final average relative score on the Hacker's Delight benchmark. While all models improve with regards to the initial proposal distribution based on uniform sampling, the model conditioning on program features reach better performances.

## 4.2 EXISTING PROGRAMS

In order to have a larger corpus than the twenty-five programs initially present in "Hacker's Delight", we generate various starting points for each optimization. This is accomplished by running Stoke with a cost function where $\omega_p = 0$ in (1), and keeping only the correct programs. Duplicate programs are filtered out. This allows us to create a larger dataset from which to learn. Examples of these programs at different level of optimization can be found in Appendix D.

We divide this augmented Hacker's Delight dataset into two sets. All the programs corresponding to even-numbered tasks are assigned to the first set, which we use for training. The programs corresponding to odd-numbered tasks are kept for separate evaluation, so as to evaluate the generalisation of our learnt proposal distribution.

The optimization process is visible in Figure 2, which shows a clear decrease of the training loss and testing loss for both models. While simply using stochastic super-optimization allows to discover programs 40% more efficient on average, using a tuned proposal distribution yield even larger improvements, bringing the improvements up to 60%, as can be seen in Table2. Due to the similarity between the different tasks, conditioning on the program features does not bring any significant improvements.

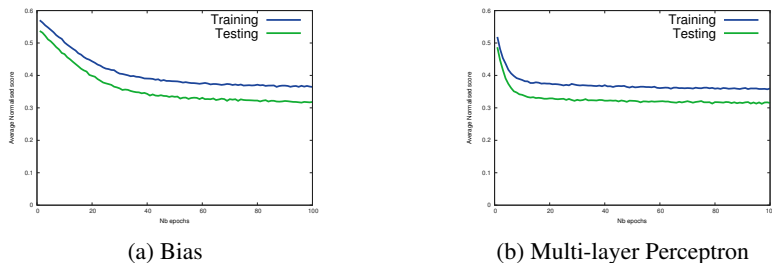

(a) Bias                    (b) Multi-layer Perceptron

Figure 2: Proposal distribution training. All models learn to improve the performance of the stochastic optimization. Because the tasks are different between the training and testing dataset, the values between datasets can't directly be compared as some tasks have more opportunity for optimization. It can however be noted that improvements on the training dataset generalise to the unseen tasks.

In addition, to clearly demonstrate the practical consequences of our learning, we present in Figure 3 a superposition of score traces, sampled from the optimization of a program of the test set. Figure 3a corresponds to our initialisation, an uniform distribution as was used in the work of Schkufza et al. (2013). Figure 3d corresponds to our optimized version. It can be observed that, while the uniform proposal distribution was successfully decreasing the cost of the program, our learnt proposal distribution manages to achieve lower scores in a more robust manner and in less iterations. Even using only 100 iterations (Figure 3e), the learned model outperforms the uniform proposal distribution with 400 iterations (Figure 3c).

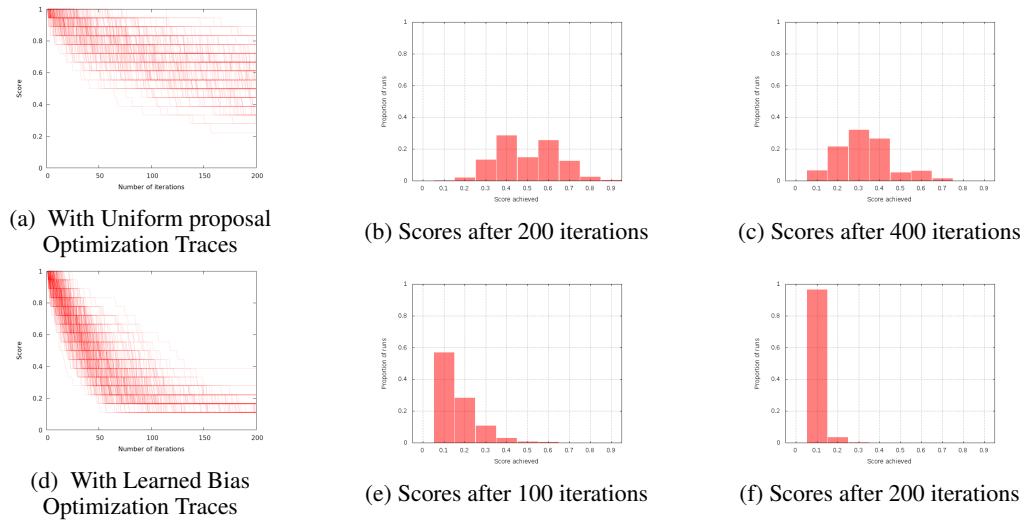

(a) With Uniform proposal Optimization Traces

(b) Scores after 200 iterations

(c) Scores after 400 iterations

(d) With Learned Bias Optimization Traces

(e) Scores after 100 iterations

(f) Scores after 200 iterations

Figure 3: Distribution of the improvement achieved when optimising a training sample from the Hacker's Delight dataset. The first column represent the evolution of the score during the optimization. The other columns represent the distribution of scores after a given number of iterations. (a) to (c) correspond to the uniform proposal distribution, (d) to (f) correspond to the learned bias.

## 4.3 AUTOMATICALLY GENERATED PROGRAMS

While the previous experiments shows promising results on a set of programs of interest, the limited diversity of programs might have made the task too simple, as evidenced by the good performance of a blind model. Indeed, despite the data augmentation, only 25 different tasks were present, all variations of the same programs task having the same optimum.

To evaluate our performance on a more challenging problem, we automatically synthesize a larger dataset of programs. Our methods to do so consists in running Stoke repeatedly with a constant cost function, for a large number of iterations. This leads to a fully random walk as every proposed programs will have the same cost, leading to a 50% chance of acceptance. We generate 600 of these programs, 300 that we use as a training set for the optimizer to learn over and 300 that we keep as a test set.

The performance achieved on this more complex dataset is shown in Figure 4 and Table 4.

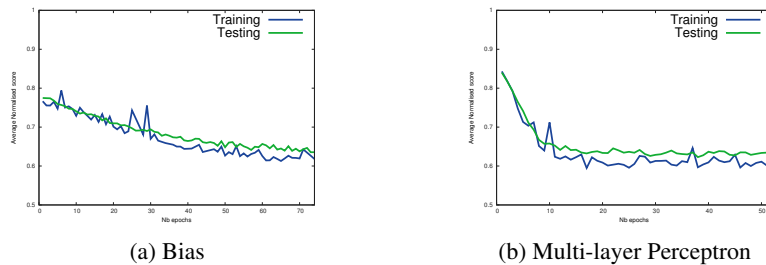

(a) Bias

(b) Multi-layer Perceptron

Figure 4: Training of the proposal distribution on the automatically generated benchmark.

| Proposal distribution | MCMC iterations throughput |
|---|---|
| Uniform | 60 000 /second |
| Categorical | 20 000 /second |

Table 3: Throughput of the proposal distribution estimated by timing MCMC for 10000 iterations

| Model | Training | Test |
|---|---|---|
| Uniform | 76.63% | 78.15 % |
| Bias | 61.81% | 63.56% |
| MLP | 60.13% | 62.27% |

Table 4: Final average relative score. The MLP conditioning on the features of the program perform better than the simple bias. Even the unconditioned bias performs significantly better than the Uniform proposal distribution.

## 5 CONCLUSION

Within this paper, we have formulated the problem of optimizing the performance of a stochastic super-optimizer as a Machine Learning problem. We demonstrated that learning the proposal distribution of a MCMC sampler was feasible and lead to faster and higher quality improvements. Our approach is not limited to stochastic superoptimization and could be applied to other stochastic search problems.

It is interesting to compare our method to the synthesis-style approaches that have been appearing recently in the Deep Learning community (Graves et al., 2014) that aim at learning algorithms directly using differentiable representations of programs. We find that the stochastic search-based approach yields a significant advantage compared to those types of approaches, as the resulting program can be run independently from the Neural Network that was used to discover them.

Several improvements are possible to the presented methods. In mature domains such as Computer Vision, the representations of objects of interests have been widely studied and as a result are successful at capturing the information of each sample. In the domains of programs, obtaining informative representations remains a challenge. Our proposed approach ignores part of the structure of the program, notably temporal, due to the limited amount of existing data. The synthetic data having no structure, it wouldn't be suitable to learn those representations from it. Gathering a larger dataset of frequently used programs so as to measure more accurately the practical performance of those methods seems the evident next step for the task of program synthesis.

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

# A  HYPERPARAMETERS

## A.1  ARCHITECTURES

The output size of 9 corresponds to the types of move. The output size of 2903 correspond to the number of possible instructions that Stoke can use during a rewrite. This is smaller that the 3874 that are possible to find in an original program.

| Outputs | Bias (9) SoftMax | Bias (2903) SoftMax |
|---|---|---|

Table 5: Architecture of the Bias

| Embedding | Linear (3874 → 100) + ReLU Linear (100 → 300) + ReLU Linear (300 → 300) + ReLU | |
|---|---|---|
| Outputs | Linear (300 → 9) SoftMax | Linear (300 → 2903) SoftMax |

Table 6: Architecture of the Multi Layer Perceptron

## A.2  TRAINING PARAMETERS

All of our models are trained using the Adam (Kingma & Ba, 2015) optimizer, with its default hyper-parameters $\beta_1 = 0.9$, $\beta_2 = 0.999$, $\epsilon = 10^{-8}$. We use minibatches of size 32.

The learning rate were tuned by observing the evolution of the loss on the training datasets for the first iterations. The picked values are given in Table 7. Those learning rates are divided by the size of the minibatches.

| | Hacker's Delight | Synthetic |
|---|---|---|
| Bias | 1 | 10 |
| MLP | 0.01 | 0.1 |

Table 7: Values of the Learning rate used.

# B  STRUCTURE OF THE PROPOSAL DISTRIBUTION

The sampling process of a move is a hierarchy of sampling step. The easiest way to represent it is as a generative model for the program transformations. Depending on what type of move is sampled, different series of sampling steps have to be performed. For a given move, all the probabilities are sampled independently so the probability of proposing the move is the product of the probability of picking each of the sampling steps. The generative model is defined in Figure 5. It is going to be parameterized by the the parameters of each specific probability distribution it samples from. The default Stoke version uses uniform probabilities over all of those elementary distributions.

```
1   def proposal(current_program):
2       move_type = sample(categorical(all_move_type))
3       if move_type == 1: % Add empty Instruction
4           pos = sample(categorical(all_positions(current_program)))
5           return (ADD_NOP, pos)
6
7       if move_type == 2: % Delete an Instruction
8           pos = sample(categorical(all_positions(current_program)))
9           return (DELETE, pos)
10
11      if move_type == 3: % Instruction Transform
12          pos = sample(categorical(all_positions(current_program)))
13          instr = sample(categorical(set_of_all_instructions))
14          arity = nb_args(instr)
15          for i = 1, arity:
16              possible_args = possible_arguments(instr, i)
17              % get one of the arguments that can be used as i-th
18              % argument for the instruction 'instr'.
19              operands[i] = sample(categorical(possible_args))
20          return (TRANSFORM, pos, instr, operands)
21
22      if move_type == 4: % Opcode Transform
23          pos = sample(categorical(all_positions(current_program)))
24          args = arguments_at(current_program, pos)
25          instr = sample(categorical(possible_instruction(args)))
26          % get an instruction compatible with the arguments
27          % that are in the program at line pos.
28          return(OPCODE_TRANSFORM, pos, instr)
29
30      if move_type == 5: % Opcode Width Transform
31          pos = sample(categorical(all_positions(current_program))
32          curr_instr = instruction_at(current_program, pos)
33          instr = sample(categorical(same_memonic_instr(curr_instr))
34          % get one instruction with the same memonic that the
35          % instruction 'curr_instr'.
36          return (OPCODE_TRANSFORM, pos, instr)
37
38      if move_type == 6: % Operand transform
39          pos = sample(categorical(all_positions(current-program))
40          curr_instr = instruction_at(current_program, pos)
41          arg_to_mod = sample(categorical(args(curr_instr)))
42          possible_args = possible_arguments(curr_instr, arg_to_mod)
43          new_operand = sample(categorical(possible_args))
44          return (OPERAND_TRANSFORM, pos, arg_to_mod, new_operand)
45
46      if move_type == 7: % Local swap transform
47          block_idx = sample(categorical(all_blocks(current_program)))
48          possible_pos = pos_in_block(current_program, block_idx)
49          pos_1 = sample(categorical(possible_pos))
50          pos_2 = sample(categorical(possible_pos))
51          return (SWAP, pos_1, pos_2)
52
53      if move_type == 8: % Global swap transform
54          pos_1 = sample(categorical(all_positions(current_program)))
55          pos_2 = sample(categorical(all_positions(current_program)))
56          return (SWAP, pos_1, pos_2)
57
58      if move_type == 9: % Rotate transform
59          pos_1 = sample(categorical(all_positions(current_program)))
60          pos_2 = sample(categorical(all_positions(current_program)))
61          return (ROTATE, pos_1, pos_2)
```

Figure 5: Generative Model of a Transformation.

## C  HACKER'S DELIGHT TASKS

The 25 tasks of the Hacker's delight Warren (2002) datasets are the following:

1. Turn off the right-most one bit
2. Test whether an unsigned integer is of the form $2^(n-1)$
3. Isolate the right-most one bit
4. Form a mask that identifies right-most one bit and trailing zeros
5. Right propagate right-most one bit
6. Turn on the right-most zero bit in a word
7. Isolate the right-most zero bit
8. Form a mask that identifies trailing zeros
9. Absolute value function
10. Test if the number of leading zeros of two words are the same
11. Test if the number of leading zeros of a word is strictly less than of another work
12. Test if the number of leading zeros of a word is less than of another work
13. Sign Function
14. Floor of average of two integers without overflowing
15. Ceil of average of two integers without overflowing
16. Compute max of two integers
17. Turn off the right-most contiguous string of one bits
18. Determine if an integer is a power of two
19. Exchanging two fields of the same integer according to some input
20. Next higher unsigned number with same number of one bits
21. Cycling through 3 values
22. Compute parity
23. Counting number of bits
24. Round up to next highest power of two
25. Compute higher order half of product of x and y

Reference implementation of those programs were obtained from the examples directory of the stoke repository (Churchill et al., 2016).

## D    EXAMPLES OF HACKER'S DELIGHT OPTIMISATION

The first task of the Hacker's Delight corpus consists in turning off the right-most one bit of a register.

When compiling the code in Listing 6a, `llvm` generates the code shown in Listing 6b. A typical example of an equivalent version of the same program obtained by the data-augmentation procedure is shown in Listing 6c. Listing 6d contains the optimal version of this program.

Note that such optimization are already feasible using the stoke system of Schkufza et al. (2013).

```
1   pushq %rbp
2   movq %rsp , %rbp
3   movl %edi , −0x4(%rbp )
4   movl −0x4(%rbp ) , %edi
5   subl $0x1 , %edi
6   movl %edi , −0x8(%rbp )
7   movl −0x4(%rbp ) , %edi
8   andl −0x8(%rbp ) , %edi
9   movl %edi , %eax
10  popq %rbp
11  retq
12  nop
13  nop
14  nop
```

```
1   #include <stdint.h>
2
3   int32_t p01(int32_t x) {
4       int32_t o1 = x − 1;
5       return x & o1;
6   }
```

(a) Source.

(b) Optimization starting point.

```
1   blsrl %edi , %esi
2   sets %ch
3   xorq %rax , %rax
4   sarb $0x2 , %ch
5   rorw $0x1 , %di
6   subb $0x3 , %dil
7   mull %ebp
8   subb %ch , %dh
9   rcrb $0x1 , %dil
10  cmovbel %esi , %eax
11  retq
```

```
1   blsrl %edi , %eax
2   retq
```

(c) Alternative equivalent program.

(d) Optimal solution.

Figure 6: Program at different stage of the optimization.

