# Peer review of "Learning to superoptimize programs"

_ICLR 2017 — accepted_

[Public Comment · Tara N Sainath · 07 Nov 2016]
**ICLR Paper Format**

Dear Authors,

Please resubmit your paper in the ICLR 2017 format with the correct font for your submission to be considered. Thank you!

[Official Review · AnonReviewer4 · rating 7 · confidence 4 · 14 Dec 2016]
**Interesting ideas, not sure it belongs @ ICLR**

Two things I really liked about this paper:
1. The whole idea of having a data-dependent proposal distribution for MCMC. I wasn't familiar with this, although it apparently was previously published. I went back: the (Zhu, 2000) paper was unreadable. The (Jampani, 2014) paper on informed sampling was good. So, perhaps this isn't a good reason for accepting to ICLR.

2. The results are quite impressive. The rough rule-of-thumb is that optimization can help you speed up code by 10%. The standard MCMC results presented on the paper on randomly-generated programs roughly matches this (15%). The fact that the proposed algorithm get ~33% speedup is quite surprising, and worth publishing.

The argument against accepting this paper is that it doesn't match the goals of ICLR. I don't go to ICLR to hear about generic machine learning papers (we have NIPS and ICML for that). Instead, I go to learn about how to automatically represent data and models. Now, maybe this paper talks about how to represent (generated) programs, so it tangentially lives under the umbrella of ICLR. But it will compete against more relevant papers in the conference -- it may just be a poster. Sending this to a programming language conference may have more eventual impact.

Nonetheless, I give this paper an "accept", because I learned something valuable and the results are very good.

[Reviewer Comment · AnonReviewer2 · 16 Dec 2016]
**Pinging authors to answer the posted questions.**

Answers to the questions posted by reviewers would help for a more high-quality review. Thanks.

[Official Review · AnonReviewer2 · rating 6 · confidence 5 · 17 Dec 2016]

This work builds on top of STOKE (Schkufza et al., 2013), which is a superoptimization engine for program binaries. It works by starting with an existing program, and proposing modifications to it according to a proposal distribution. Proposals are accepted according to the Metropolis-Hastings criteria. The acceptance criteria takes into account the correctness of the program, and performance of the new program. Thus, the MCMC process is likely to converge to correct programs with high performance. Typically, the proposal distribution is fixed. The contribution of this work is to learn the proposal distribution as a function of the features of the program (bag of words of all the opcodes in the program). The experiments compare with the baselines of uniform proposal distribution, and a baseline where one just learns the weights of the proposal distribution but without conditioning on the features of the program. The evaluation shows that the proposed method has slightly better performance than the compared baselines.

The significance of this work at ICLR seems to be quite low., both because this is not a progress in learning representations, but a straightforward application of neural networks and REINFORCE to yet another task which has non-differentiable components. The task itself (superoptimization) is not of significant interest to ICLR readers/attendees. A conference like AAAI/UAI seem a better fit for this work.

The proposed method is seemingly novel. Typical MCMC-based synthesis methods are lacking due to their being no learning components in them. However, to make this work compelling, the authors should consider demonstrating the proposed method in other synthesis tasks, or even more generally, other tasks where MH-MCMC is used, and a learnt proposal distribution can be beneficial. Superoptimization alone (esp with small improvements over baselines) is not compelling enough.

It is also not clear if there is any significant representation learning is going on. Since a BoW feature is used to represent the programs, the neural network cannot possibly learn anything more than just correlations between presence of opcodes and good moves. Such a model cannot possibly understand the program semantics in any way. It would have been a more interesting contribution if the authors had used a model (such as Tree-LSTM) which attempts to learn the semantics the program. The quite naive method of learning makes this paper not a favorable candidate for acceptance.

[Official Review · AnonReviewer3 · rating 8 · confidence 4 · 17 Dec 2016]

This is an interesting and pleasant paper on superoptimization, that extends the  problem approached by the stochastic search STOKE to a learned stochastic search, where the STOKE proposals are the output of a neural network which takes some program embedding as an input. The authors then use REINFORCE to learn an MCMC scheme with the objective of minimizing the final program cost.

The writing is clear and results highlight the efficacy of the method.

comments / questions:
- Am I correct in understanding that of the entire stochastic computation graph, only the features->proposal part is learned. The rest is still effectively the stoke MCMC scheme? Does that imply that the 'uniform' model is effectively Stoke and is your baseline (this should probably be made explicit )

- Did the authors consider learning the features instead of using out of the box features (could be difficult given the relatively small amount of data - the feature extractor might not generalize).

- In a different context, 'Markov Chain Monte Carlo and Variational Inference:Bridging the Gap' by Salimans et al. suggests considering a MCMC scheme as a stochastic computation graph and optimizing using a variational (i.e. RL) criterion. The problem is different, it uses HMC instead of MCMC, but it might be worth citing as a similar approach to 'meta-optimized' MCMC algorithms.

[Public Comment · Rudy R Bunel · 17 Jan 2017]
**Uploaded new version**

We thank the reviewers for their helpful feedback. We uploaded a new version of
the paper. The elements that have changed are:

Clarity of experiments:
- The complexity of each model in term of number of parameters is given in Table 1 and the hyperparameters / architecture were added as Appendix A.
- Addition of a better representation of the consequences of a learned bias (Figure 3). We compare Stoke (uniform distribution)  running for (200,400) iterations with the learned optimizer running for (100, 200) iterations (at training time, the horizon is set at 200 iterations). Even with four times less iterations, a better distribution of the score is achieved.
- The numbers for the impact on throughput of the learned proposal distribution have been added (Table 3). This measures not just the time for the proposal (which we gave as answer to the question of AnonReviewer2) but the whole time of an MCMC iterations which represent more accurately the throughput of the algorithm.

Explanations:
- We added the citation that AnonReviewer3 suggested (Section 2 - Related Works).
- We added our argument that superoptimization provides a good benchmark to estimate representation of programs to the introduction (Section 1 - Introduction, 1st paragraph).
- Clarified throughout the paper that the "Uniform" model is Stoke (Section 3.2.2-Parametrization and Section 4.1-Models, Table 1).
- Fixed some typos

[Final Decision · Program Chairs · 06 Feb 2017]
**ICLR committee final decision**

This paper applies REINFORCE to learn MCMC proposals within the existing STOKE scheme for super-optimization. It's a neat paper, with interesting results.
 
 It's not clear whether interesting representations are learned, and the algorithms are not really new. However, it's a neat piece or work, that some ICLR reviewers found interesting, and could inspire more representation learning work in this area.